# Sexually transmitted infections among key populations in India: A protocol for systematic review

**Mihir Bhatta**[1], **Agniva Majumdar**[1], **Utsha Ghosh**[1], **Piyali Ghosh**[1], **Papiya Banerji**[1], **Santhakumar Aridoss**[2], **Abhisek Royal**[3], **Subrata Biswas**[1]*, **Bhumika Tumkur Venkatesh**[4], **Rajatsuvra Adhikary**[3], **Shanta Dutta**[5]

1 Division of Virology, ICMR-National Institute of Cholera and Enteric Diseases, Kolkata, India, 2 Division of Computing and Information Science, ICMR-National Institute of Epidemiology, Chennai, India, 3 World Health Organization, New Delhi, India, 4 Campbell Collaboration, South Asia, New Delhi, India, 5 Division of Bacteriology, ICMR-National Institute of Cholera and Enteric Diseases, Kolkata, India

* sai_brata@yahoo.com

**Data Availability Statement:** No datasets were generated or analysed during the current study. All relevant data from this study will be made available upon study completion.

## Abstract

### Background

Sexually transmitted infections (STIs) are one of the leading causes of health, and economic burdens in the developing world, leading to considerable morbidity, mortality, and stigma. The incidence and prevalence of the four curable STIs viz. syphilis, gonorrhoea, chlamydia, and trichomoniasis vary remarkably across different geographical locations. In India, the prevalence of four curable STI among general populations is in between 0 to 3.9 percent. However, it is assumed that STI prevalence is much higher among subpopulations practicing high-risk behaviour. Like men who have sex with men (MSM), transgender (TG), injecting drug users (IDU), and female sex workers (FSW).

### Objectives

In the present circumstances, a systematic review is necessary to integrate the available data from previously published peer-reviewed articles and published reports from several competent authorities to provide the prevalence and geographical distribution of the four curable STIs among the key population of India.

### Methods

All available articles will be retrieved from PubMed, Google Scholar, Cochrane database, Scopus, Science Direct, and the Global Health network using the appropriate search terms. The data will be extracted through data extraction form as per PICOS (population, intervention, comparison, outcome, study design) framework. Risk of bias and quality assessment will be performed according to the situation with the help of available conventional protocol.

### Discussion

The future systematic review, generated from the present protocol, may provide evidence of the prevalence and geographical distribution of the four curable STIs among the key

**Funding:** Fund for this research is provided by WHO-India (Reg. No. 2022/1250989). Moreover, the funders had no role in the study design, data collection, analysis, the decision to publish, or the preparation of the manuscript.

**Competing interests:** he authors have declared that no competing interests exist.

population of India. We hope that the findings of the future systematic review will strengthen the existing surveillance system in India, to determine the above-mention STIs prevalence among key populations in India.

**Protocol registration number:** International Prospective Register for Systematic Reviews (PROSPERO) number CRD42022346470.

## Introduction

According to a recent estimate by World Health Organization (WHO), approximately three hundred seventy million new sexually transmitted infections (STIs) occur each year around the world, and almost half of the infected individuals are young in their age [1]. For the suitable planning and execution of sexually transmitted disease (STD) control strategies, a proper understanding of the patterns of STDs, prevailing at different geographic pockets is required [2]. Individuals suffering from STIs are not only susceptible to getting infected with HIV but also play an important role in the transmission of STIs and HIV to others [3, 4].

Recent research indicates synergy between bacterial STIs and HIV transmission and acquisition and in turn enhances transmission and or acquisitions of HIV among high-risk groups (HRGs) who have sex with men (MSM), transgender (TG), injecting drug users (IDU), and female sex workers (FSW) [2]. According to WHO [1], more than thirty different bacteria, viruses, and parasites are transmitted sexually. Among them, eight are selected as the prominent causal agent of sexually transmitted diseases. Between these eight, four are curable viz. syphilis, gonorrhoea, chlamydia, and trichomoniasis [5]. The other four are incurable viz. hepatitis B, herpes simplex virus (HSV or herpes), HIV, and human papillomavirus (HPV) [5]. STIs are spread through unprotected sexual contact, which includes vaginal, anal, and oral routes. Certain organisms are also transmitted through maternal lineage during pregnancy, childbirth, and breastfeeding. Mother-to-child transmission of STIs has effects like stillbirth, neonatal death, low-birth weight and prematurity, sepsis, pneumonia, neonatal conjunctivitis, and congenital deformities [6]. People with STIs rarely show the symptoms of the disease. General symptoms of STIs, are abdominal pain, urethral discharge or burning in men, vaginal discharge, and genital ulcers. STIs remain a major public health challenge for people belonging to high-risk groups (for HIV) around the world [6]. During the last twenty years, the National AIDS Control Organization (NACO) has undertaken the prevention of STIs as one of its key strategies in India [7]. NACO through its network of more than eleven hundred designated clinics branded as Suraksha Clinics, located mostly at the district-level government healthcare facilities provides free sexual and reproductive health services based on syndromic case management through trained counsellors, expert paramedics, and experienced medical personnel [8].

NACO is the authority for the management of STI to ensure consistency of service across all facilities, over the country. The recently released strategy document of NACP Phase-V reinforces the STI component in terms of the elimination of vertical transmission of HIV and syphilis [8]. In India, the prevalence of four curable STIs among general populations is less than four percent, but the STI burden is probably much higher among subpopulations practicing high-risk behaviour like MSM, TGs, IDU, and FSWs. There is limited literature on STI prevalence among key populations across India but of course there are still relevant studies available these data have not been integrated to depict the overall spatial and temporal trends of STI infections among various key populations [9]. In the present circumstances, a systematic review (or if possible and meta-analysis) is needed to integrate the available data from

previously published peer-reviewed articles and published reports from several competent authorities [9]. The present protocol for a systematic review is set against this background and intended to include an intensive consultation with various experts, program managers, and representatives of key populations to recognize the present prevalence and geographic distribution of STIs among key populations in India. The present findings would be vital for enlightening STI status among high-risk group people and designing evidence-based strategies and programs for STI prevention in high-risk group people in India.

## Objective

The objectives of the present study are as follows:

A. Synthesize evidence on the prevalence of four curable STIs viz. syphilis, gonorrhoea, chlamydia, and Trichomoniasis (caused by *Treponema pallidum*, *Neisseria gonorrhoeae* (NG), *Chlamydia trachomatis* (CT), and *Trichomonas vaginalis* (TV) respectively) among FSW, MSM, IDU and H/TG populations in India.

B. Perform a Systematic Review (and Meta-analysis, upon the availability of necessary data) on existing evidence of the prevalence (pooled prevalence, in case of meta-analysis) as well as the geographical distribution of four curable STIs among key populations in India from previously published articles.

## Materials and methods

A study protocol is developed following Preferred Reporting Items for Systematic Reviews and Meta-Analysis (PRISMA) guidelines [10]. The protocol is registered in the PROSPERO [11], International Prospective Register of Systematic Reviews with the registration number CRD42022357425 [12].

### Review questions

i. What is the current prevalence of four curable sexually transmitted infections such are, syphilis, gonorrhea, chlamydia and Trichomoniasis (caused by *Treponema pallidum*, *Neisseria gonorrhoeae* (NG), *Chlamydia trachomatis* (CT), and *Trichomonas vaginalis* (TV) respectively), among the key populations (FSW, MSM, H/TG, and IDU) in India?

ii. What is the distribution of four curable sexually transmitted infections at the different geographical locations/ representations across India?

### Inclusion criteria

PICOS (population, intervention, comparison, outcome, study design) framework is used for defining systematic review questions according to the method described by the Cochrane Handbook for Systematic Reviews of Interventions version 6.3 [13]. Any article published in peer-reviewed journals, any reports by the government or authorized and competent non-governmental agencies during the accepted timeline (i.e., January 2001 to December 2022).

**Population.**

i. The population belongs to the key population, which are FSW, MSM, H/TG and IDU in India (as per NACO Case definition).

ii. Adults and young people aged 18 years and over.

iii. Women, men, and transgender.

**Intervention.** Not relevant to this review of observational studies.

**Comparison group.** Not relevant to this review.

**Outcome.** Individual and/or cumulative pooled prevalence of four curable sexually transmitted infections are, *Treponema pallidum*, *Neisseria gonorrhoeae* (NG), *Chlamydia trachomatis* (CT) and *Trichomonas vaginalis* (TV) among the key populations (FSW, MSM H/TG and IDU) across different geographical regions in India.

**Study design.** Data will be extracted through a previously prepared *Data Extraction Form* which includes different levels of data accumulation to obtain a general picture of four curable STIs among the key population belonging to different geographical regions in India, from previously published articles divided into cross-sectional or baseline studies on different key population or cohort studies and different state/ central government reports regarding these four curable STIs among FSW, MSM H/TG and IDU. After the completion of data extraction, data will be analysed and a cumulative report of the systematic review will be generated.

## Exclusion criteria

i. Countries other than India;

ii. Serological and syndromic studies, sampling other than these four curable STIs;

iii. The study population does not belong to FSW, MSM, H/TG, and IDU;

iv. Any article does not contain data on either/any four curable STIs among FSW, MSM, H/ TG, and IDU;

v. Articles published not within the accepted timeline;

vi. Participants aged less than 18 years;

vii. Commentaries and editorials.

## Search strategies and selection process

**Electronics databases.** The following databases will be searched from January 2001 to December 2022 for published articles in English language. The search will be updated before initiating a statistical analysis.

a. MEDLINE

b. Cochrane Library

c. Psychinfo

d. Science Direct

e. Scopus

f. EMBASE

g. Google Scholar and

h. PUBMED

**Grey literature.** An extensive search will be carried out through the following reports relevant to the study.

a. Annual reports of NACO

b. Reports by SANKALAK (NACO)

c. Annual reports of UNAIDS

d. Annual reports of WHO

e. Annual reports of UNDP

f. Annual reports of ICMR

g. Ganga Social Foundation

h. West Bengal State AIDS Prevention and Control Society

i. Tamil Nadu State AIDS Prevention Control Society

j. Annual reports of Delhi State AIDS Control Society

k. State reports of Kerala, Rajasthan

l. MGVS annual report

m. KSAPS annual report

Inclusion of studies from Grey literature will be carried out after being checked through a quality assessment tool (Axis Tool) [14].

## Search terms

Search terms are provided in S1 File

## Additional searches

i. Reference lists: if retrieved publications will be include source references for potential studies about the prevalence of four curable STIs on any members of the Key population, the originals will be retrieved;

ii. Experts in the field will be contacted to ask if they know of any additional publications, which will not be identified by the search strategy.

## De-duplication

i. Mendeley [15], a bibliographic application will be used for reference management.

ii. The following rules will be used to remove duplicate hits from the database

iii. Title, or various combinations of the author, year, secondary title, volume, issue, and pages will be compared through the 'de-duplication' process

iv. The full records of suspected duplicates will be compared visually

v. Duplicate entries will be saved in a separate MS Word file

### Selection of eligible studies

Titles and abstracts of articles selected through the search strategy will be screened by two reviewers independently, applying the inclusion and exclusion criteria. Any article selected as being probably qualified will be taken for the full text review. Where no abstract will be available electronically, and eligibility could not be judged from the title alone, the full text of the article will be retrieved and screened. The abstracts of articles identified through additional searches will be reviewed in the same manner as those identified through database searches. Data will be extracted by data extraction form.

### Strategy for data synthesis

The data will be extracted from full-text published articles according to the Preferred Reporting Items for Systematic Reviews and Meta-Analysis (PRISMA). With the help of Epi Info— (Ver. 6.0) generated modified data extraction form [16].

### Assessment of the methodological quality

Evaluation of articles through name, abstract, and entire text of the selected articles will be performed prior to the addition of it in the ultimate analysis. Assessment will be performed with the help of a modified Newcastle—Ottawa Quality Assessment Scale [17].

### Publication bias analysis

To evaluate publication bias Egger's [18] and Begg and Mazumdar's [19] assessments will be used along with the Funnel diagram.

### Descriptive analysis

The prevalence (pooled prevalence in case of meta-analysis) estimation will be done from each study on above mentioned four curable sexually transmitted infections among the key populations in India.

### Strength and limitations of the study

Like any systematic review, the present study will also be restricted by the comprehensiveness of the published articles and whether workers published their study in open accessed and peer-reviewed journals as well as available reports from various agencies. Moreover, future included articles that will be included as per the current protocol may have to contain surveys that accumulate biological data leaving an approach for further studies.

### Expected outcomes

The future systematic review, which will be generated from the present protocol, may provide evidence on the current prevalence (pooled prevalence in case of meta-analysis) and present geographical distribution of four curable STIs viz. Syphilis, Gonorrhoea, Chlamydia, and Trichomoniasis among key populations in India.

## Discussions

The future systematic review, which will be generated from the present protocol, may provide evidence of the prevalence and geographical distribution of the four curable STIs among the key population of India. The findings of the future systematic review will strengthen the existing surveillance system under NACP-V (fifth phase of National AIDS Control Programme

under the supervision of NACO, Government of India. This phase is renamed as National AIDS & STI Control Programme), to determine the above-mentioned STIs prevalence among key populations in India. The present protocol might be handy to conduct a systemic review of the prevalence and geographical distribution of the four above-mentioned bacterial STIs among the general population in India.

## Supporting information

**S1 Checklist. PRISMA-P (Preferred Reporting Items for Systematic review and Meta-Analysis Protocols) 2015 checklist: Recommended items to address in a systematic review protocol**∗.
(PDF)

**S1 File. Search terms and strategy.**
(DOCX)

**S2 File. Data extraction form.**
(DOCX)

## Author Contributions

**Conceptualization:** Mihir Bhatta, Subrata Biswas, Bhumika Tumkur Venkatesh, Rajatsuvra Adhikary, Shanta Dutta.

**Data curation:** Mihir Bhatta, Utsha Ghosh, Piyali Ghosh, Papiya Banerji, Subrata Biswas.

**Formal analysis:** Utsha Ghosh, Piyali Ghosh, Papiya Banerji.

**Funding acquisition:** Abhisek Royal, Rajatsuvra Adhikary, Shanta Dutta.

**Investigation:** Mihir Bhatta, Utsha Ghosh, Piyali Ghosh, Papiya Banerji, Santhakumar Aridoss, Subrata Biswas.

**Methodology:** Mihir Bhatta, Utsha Ghosh, Papiya Banerji, Santhakumar Aridoss, Subrata Biswas, Shanta Dutta.

**Project administration:** Mihir Bhatta, Subrata Biswas, Rajatsuvra Adhikary, Shanta Dutta.

**Resources:** Agniva Majumdar, Abhisek Royal, Rajatsuvra Adhikary, Shanta Dutta.

**Supervision:** Mihir Bhatta, Agniva Majumdar, Papiya Banerji, Santhakumar Aridoss, Subrata Biswas, Bhumika Tumkur Venkatesh, Rajatsuvra Adhikary, Shanta Dutta.

**Validation:** Santhakumar Aridoss.

**Writing – original draft:** Mihir Bhatta, Utsha Ghosh, Piyali Ghosh, Papiya Banerji, Subrata Biswas.

**Writing – review & editing:** Mihir Bhatta, Agniva Majumdar, Utsha Ghosh, Piyali Ghosh, Papiya Banerji, Santhakumar Aridoss, Subrata Biswas, Shanta Dutta.

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
