## [Decision Letter · Decision Letter 0]

7 Feb 2023

PONE-D-22-31768

Sexually Transmitted Infections among Key Populations in India: A Protocol for Systematic Review

PLOS ONE

Dear Dr. Biswas,

Thank you for submitting your manuscript to PLOS ONE. After careful consideration, we feel that it has merit but does not fully meet PLOS ONE’s publication criteria as it currently stands. Therefore, we invite you to submit a revised version of the manuscript that addresses the points raised during the review process.

We look forward to receiving your revised manuscript.

Kind regards,

Addisu Melese Dagnaw, MSc

Academic Editor

PLOS ONE

Journal Requirements:

   "Fund for this research is provided by WHO-India (Reg. No. 2022/1250989)."

Reviewers' comments:

Reviewer's Responses to Questions

**Comments to the Author**

1. Does the manuscript provide a valid rationale for the proposed study, with clearly identified and justified research questions?

Reviewer #1: Yes

Reviewer #2: Yes

Reviewer #3: No

2. Is the protocol technically sound and planned in a manner that will lead to a meaningful outcome and allow testing the stated hypotheses?

Reviewer #1: Yes

Reviewer #2: Yes

Reviewer #3: No

3. Is the methodology feasible and described in sufficient detail to allow the work to be replicable?

Reviewer #1: Yes

Reviewer #2: Yes

Reviewer #3: No

4. Have the authors described where all data underlying the findings will be made available when the study is complete?

Reviewer #1: Yes

Reviewer #2: Yes

Reviewer #3: No

5. Is the manuscript presented in an intelligible fashion and written in standard English?

Reviewer #1: Yes

Reviewer #2: Yes

Reviewer #3: No

6. Review Comments to the Author

You may also provide optional suggestions and comments to authors that they might find helpful in planning their study.

Reviewer #1: This protocol is well written, with a clear objective to get a better insight in the prevalence of four major sexually transmitted infections in India. I have only some minor comments to further improve the manuscript.

Minor comments and editorials

Introduction

-In line 9 is mentioned that ‘almost half of them are much younger of age’. This is not comprehensive. Half of what? Younger than who? Please clarify and adjust this sentence.

-Line 14: In this line STIs and HIV is written. Hiv infection actually is an STI. Please adjust.

-Line 29: please remove the comma between ‘STIs’ and ‘are’.

-Line 44: Please add ‘probably’ before ‘much higher’, since the STI prevalences are the subject of this study.

-Line 45: It is mentioned that ‘there is limited literature’ which is indeed why this review will be performed, but of course there are still relevant studies available. It would be good to already refer to some major studies here.

-In line 48 reference 9 is given. It suggests that this will refer to key populations in India but that is not the case in ref 9. Please adjust.

Objective

-Line 61: please add ‘the’ before ‘present’.

-Line 70: Please remove the hard return.

Methods

-Lines 147-154: please consider to add a ‘snowballing strategy’ (=use references within references)

-Line 181: please rewrite to ‘will be included’

Strengths and limitations

-Line 233-234: ‘moreover, future included…’: this sentence is not comprehensive; please rewrite.

Conclusions

-Line 243: Actually there are no conclusions yet. Maybe rename this paragraph ‘Discussion’? Also in the abstract?

-Line 248: ‘NACP’ should probably be ‘NACO’?

-Line 250: Please adjust to ‘systematic’

Reviewer #2: 1. Methodology

i. This review may include studies that use different methods for diagnosing STIs, such as the Nucleic Acid Amplification Test (NAAT) – the gold standard test for N. gonorrhea, C. trachomatis, and T. vaginalis – as well as those that rely on clinical examination instead of laboratory investigations.

What strategy will be utilized for this issue (as it will have an impact on the pooled prevalence; varying sensitivity specificity of diagnosis methods)? This must be mentioned in the method section.

ii. Is there any operational definition for the 'geographical region' mentioned in the protocol? This may be clearly stated as it is not clear how the data will be pooled/analyzed/presented in later stages. e. g. Does the geographical region mean physical divisions of Indian Geography ? or state-wise division or zone-wise?

iii. Though from the protocol it appears that authors will be considering only observational designs, it makes it more convenient for all readers if the types of study designs, to be included, are mentioned clearly at the outset.

2. Data extraction:

Suggestion: i. You may consider enumerating a few details of the extraction form. Some of them may be sample size, setting, urban/rural, duration of the study, completeness of follow-up, enumeration of variables that are adjusted in the included study, and other factors that may influence the validity of the results. Also, administrative details like: study author, year, place, published/unpublished status

3. Data analysis:

i. The software to be used doing the meta-analysis (Individual and/or cumulative pooled prevalence )may be mentioned.

ii. Suggestion: Consideration for subgroup analysis e.g. based on categories chosen: MSM, TGs, IV drug users, and FSWs; Based on the state's population and size; per capita. income; state categorization based on the health index (NITI Aayog)

iii. Few details on how the assessment of heterogeneity will be done needs to mentioned.

iv. Do the authors plan to conduct sensitivity analyses in cases of substantial heterogeneity?

v. How the missing data will be dealt?

4. Abstract of the protocol:

i. There is literature from India substantiating the high STI prevalence among the sub-population considered for this review ( e.g. Syphilis prevalence, was reported to be 13% among MSM in Mumbai in a study published in 2009) hence assuming word in the abstract i.e., "assumed that STI prevalence is much higher among subpopulations practicing high-risk behavior", may be modified.

ii. Please specify the tools to be used for risk of bias, and quality assessment mentioned in the abstract. Please clarify the role of the Funnel plot.

iii. The authors have taken on an intriguing topic for this review, however, their abstract conclusion is not adequately supported. Therefore, the conclusion may be rephrased in a manner that emphasizes and highlights the unique contributions that this review makes to the existing body of literature.

Reviewer #3: Thank you for the opportunity to review this protocol. Overall, this protocol requires a major revision before it could be submitted for the consideration. This will also require an English editing before resubmission.

Specific comments:

Abstract:

- Background could be shortened.

- For objective, recommend to only include the main objective of the study rather than providing the rationale behind it, which could be part of a background section.

-Need to elaborate the methods section. What would be the inclusion criteria for the selection of articles (e.g., time period, language etc.)? What key words will be used?

Introduction:

- Page 2, Line 8: How is the information on young age relevant to the context of this study?

-Page 2, Line 13 and 16: Spell out the abbreviations when using for the first time in the text.

- Page 2, Line 30: Needs citation.

- Needs more concrete rationale/justification for the conduct of this systematic review.

- Objective could be included in the last paragraph of the introduction section.

- Objective A: Isn't synthesis part of a systematic review? In systematic review, we synthesize study findings from the relevant articles.

Materials and Methods:

- Requires overall revision of this section.

- Page 3, Line 75: Authors have specified "A study protocol (published)". This indicates that the protocol is already published. If it's already published, why is there a need of publishing the protocol again?

- What geographical divisions will be sued by the authors?

- Page 3, Line 97: What's the reason behind selecting the given timeline is not clear.

- Authors should consider writing methods section in paragraph rather than in bullet points.

- Strengths and limitations section needs to be more developed as well as paraphrased.

7. PLOS authors have the option to publish the peer review history of their article (what does this mean?). If published, this will include your full peer review and any attached files.

Reviewer #1: **Yes: **Sylvia M. Bruisten

Reviewer #2: **Yes: **Dr. Ranadip Chowdhury, Dr. Barsha Pathak

Reviewer #3: No

---

## [Author Response · Author response to Decision Letter 0]

16 Feb 2023

Reviewer #1:

This protocol is well written, with a clear objective to get a better insight in the prevalence of four major sexually transmitted infections in India. I have only some minor comments to further improve the manuscript.

Minor comments and editorials

Introduction

In line 9 is mentioned that ‘almost half of them are much younger of age’. This is not comprehensive. Half of what? Younger than who? Please clarify and adjust this sentence.

Authors’ reply: Authors fully agreed with the reviewer about this, and rephrased the requisite statement accordingly. Please, have a look on the manuscript.

Line 14: In this line STIs and HIV is written. HIV infection actually is an STI. Please adjust.

Authors’ reply: Authors fully agreed with the reviewer about this, and rephrased the requisite statement accordingly. Please, have a look on the manuscript.

Line 29: please remove the comma between ‘STIs’ and ‘are’.

Authors’ reply: Authors fully agreed with the reviewer about this, and rephrased the requisite statement accordingly. Please, have a look on the manuscript.

Line 44: Please add ‘probably’ before ‘much higher’, since the STI prevalence are the subject of this study.-Line 45: It is mentioned that ‘there is limited literature’ which is indeed why this review will be performed, but of course there are still relevant studies available. It would be good to already refer to some major studies here.

Authors’ reply: Authors fully agreed with the reviewer about this, and rephrased the requisite statement accordingly. Please, have a look on the manuscript.

In line 48 reference 9 is given. It suggests that this will refer to key populations in India but that is not the case in ref 9. Please adjust.

Authors’ reply: Authors fully agreed with the reviewer about this, and rephrased the requisite statement accordingly. Please, have a look on the manuscript.

Objective

Line 61: please add ‘the’ before ‘present’

Authors’ reply: Authors fully agreed with the reviewer about this, and rephrased the requisite statement accordingly. Please, have a look on the manuscript.

Line 70: Please remove the hard return.

Authors’ reply: Authors fully agreed with the reviewer about this, and rephrased the requisite statement accordingly. Please, have a look on the manuscript.

Methods

Lines 147-154: please consider to add a ‘snowballing strategy’ (=use references within references)

Authors’ reply: Authors fully agreed with the reviewer about this, and rephrased the requisite statement accordingly. Please, have a look on the manuscript.

Line 181: please rewrite to ‘will be included’

Authors’ reply: Authors fully agreed with the reviewer about this, and rephrased the requisite statement accordingly. Please, have a look on the manuscript.

Strengths and limitations

Line 233-234: ‘moreover, future included…’: this sentence is not comprehensive; please rewrite.

Authors’ reply: Authors fully agreed with the reviewer about this, and rephrased the requisite statement accordingly. Please, have a look on the manuscript.

Conclusions

Line 243: Actually there are no conclusions yet. Maybe rename this paragraph ‘Discussion’? Also in the abstract?

Authors’ reply: Authors fully agreed with the reviewer about this, and rephrased the requisite statement accordingly. Please, have a look on the manuscript.

Line 248: ‘NACP’ should probably be ‘NACO’?

Authors’ reply: NACP is the National AIDS Control Programme under NACO (National AIDS Control Organisation), Government of India. At present phase-V is going on which is renamed as National AIDS & STI Control Programme.

Line 250: Please adjust to ‘systematic

Authors’ reply: Authors fully agreed with the reviewer about this, and rephrased the requisite statement accordingly. Please, have a look on the manuscript.

 

Reviewer #2: 

1. Methodology

i. This review may include studies that use different methods for diagnosing STIs, such as the Nucleic Acid Amplification Test (NAAT) – the gold standard test for N. gonorrhea, C. trachomatis, and T. vaginalis – as well as those that rely on clinical examination instead of laboratory investigations.

Authors’ reply: Authors are fully agreed with the reviewer about this fact also like to mention that this is a Review protocol not the review itself, when we will able to perform the Systematic Review, we will include all the diagnostic methods performed along with Nucleic Acid Amplification Test (NAAT).

What strategy will be utilized for this issue (as it will have an impact on the pooled prevalence; varying sensitivity specificity of diagnosis methods)? This must be mentioned in the method section.

Authors’ reply: As of now we only can think about the systematic review, if the situation comes in such a way that favours us in performing Meta-analysis, then we can think about doing the pooled prevalence

ii. Is there any operational definition for the 'geographical region' mentioned in the protocol? This may be clearly stated as it is not clear how the data will be pooled/analyzed/presented in later stages. e.g. Does the geographical region mean physical divisions of Indian Geography? Or state-wise division or zone-wise?

Authors’ reply: Here, 'geographical region' is nothing but the zone-wise division of Indian political territory.

iii. Though from the protocol it appears that authors will be considering only observational designs, it makes it more convenient for all readers if the types of study designs, to be included, are mentioned clearly at the outset.

Authors’ reply: Here we like to add that we are not considering observational designs only, rather we like to add all the types of study designs (as available). Reviewer is requested to kindly go through the Data Extraction Form, please.

2. Data extraction:

Suggestion: i. You may consider enumerating a few details of the extraction form. Some of them may be sample size, setting, urban/rural, duration of the study, completeness of follow-up, enumeration of variables that are adjusted in the included study, and other factors that may influence the validity of the results. Also, administrative details like: study author, year, place, published/unpublished status

Authors’ reply: Thanks for you kind suggestion. We had included every point of your concern in the Data Extraction Form. 

Reviewer is requested to kindly go through the Data Extraction Form, please.

3. Data analysis:

i. The software to be used doing the meta-analysis (Individual and/or cumulative pooled prevalence) may be mentioned.

Authors’ reply: Authors are fully agreed with the reviewer about this fact also like to mention that this is a Review protocol not the review itself. As of now we only can think about the systematic review, if the situation comes in such a way that favours us in performing Meta-analysis, then we will use STATA 13.0 (we usually use STATA for analysis Funnel and Forest plot preparation along with Zotero and web portal like rayyan.ai for study data management).

ii. Suggestion: Consideration for subgroup analysis e.g. based on categories chosen: MSM, TGs, IV drug users, and FSWs; Based on the state's population and size; per capita. Income; state categorization based on the health index (NITI Aayog).

iii. Few details on how the assessment of heterogeneity will be done needs to mention.

iv. Do the authors plan to conduct sensitivity analyses in cases of substantial heterogeneity?

v. How the missing data will be dealt?

Authors’ reply: Authors are fully agreed with the reviewer about this fact also like to mention that this is a Review protocol not the review itself. As of now we only can think about the systematic review, if the situation comes in such a way that favours us in performing Meta-analysis, then we will perform subgroup analysis, assess heterogeneity. We also have a definite plan to conduct sensitivity analyses in cases of there is substantial heterogeneity. Moreover, we will deal with missing data with conventional protocol.

4. Abstract of the protocol:

i. There is literature from India substantiating the high STI prevalence among the sub-population considered for this review (e.g. Syphilis prevalence, was reported to be 13% among MSM in Mumbai in a study published in 2009) hence assuming word in the abstract i.e.," assumed that STI prevalence is much higher among subpopulations practicing high-risk behavior", may be modified.

Authors’ reply: Authors fully agreed with the reviewer about this and modified the sentence accordingly. 

ii. Please specify the tools to be used for risk of bias, and quality assessment mentioned in the abstract. Please clarify the role of the Funnel plot.

Authors’ reply: Authors fully agreed with the reviewer about this and reframed the sentence as “Risk of bias and quality assessment will be performed according to the situation with the help of available conventional protocol”.

iii. The authors have taken on an intriguing topic for this review, however, their abstract conclusion is not adequately supported. Therefore, the conclusion may be rephrased in a manner that emphasizes and highlights the unique contributions that this review makes to the existing body of literature.

Authors’ reply: Agreed with the reviewer suggestion we omitted the section Conclusion and rewrite the section as “Discussion: The future systematic review, generated from the present protocol, may provide evidence of the prevalence and geographical distribution of the four curable STIs among the key population of India. We hope that the findings of the future systematic review will strengthen the existing surveillance system in India, to determine the above-mention STIs prevalence among key populations in India”.

 

Reviewer #3: 

Thank you for the opportunity to review this protocol. Overall, this protocol requires a major revision before it could be submitted for the consideration. This will also require an English editing before resubmission.

Specific comments:

Abstract:

Background could be shortened.

For objective, recommend to only include the main objective of the study rather than providing the rationale behind it, which could be part of a background section.

Need to elaborate the methods section. What would be the inclusion criteria for the selection of articles (e.g., time period, language etc.)? What key words will be used?

Authors’ response: 

Background section is justified the present scenario as well as the rationale behind the present work.

For objective no rational is given.

We like to elaborate the method section but it is already so large (from line no 80 -256). For inclusion and exclusion criteria Reviewer is requested to go through the text from line no. 97-147. For key words Reviewer is requested to vie the search terms.

Introduction:

Page 2, Line 8: How is the information on young age relevant to the context of this study?

Authors’ response: people representing the young age group are the backbone of a society/ country. If the prevalence of STI is more among young people it will be a major threat to the steady and continuous growth of a nation.

Page 2, Line 13 and 16: Spell out the abbreviations when using for the first time in the text.

Authors’ response: when using first time in text all the abbreviations with their full form are mentioned accordingly.

Page 2, Line 30: Needs citation.

Needs more concrete rationale/justification for the conduct of this systematic review.

Objective could be included in the last paragraph of the introduction section.

Authors’ response: Citation is already mentioned as follows

6. Rowley J, Vander Hoorn S, Korenromp E, Low N, Unemo M, Abu-Raddad LJ, et al. Chlamydia, gonorrhoea, trichomoniasis and syphilis: global prevalence and incidence estimates, 2016. Bulletin of the World Health Organization. 2019;97(8):548–562P.

Objective A: Isn't synthesis part of a systematic review? In systematic review, we synthesize study findings from the relevant articles.

Authors’ response: Agreed with the reviewer comments. But this is a Review protocol not the Systematic Review itself. When we will perform the Systematic Review, such objective will be mentioned as per the conventional protocol.

Materials and Methods:

Page 3, Line 75: Authors have specified "A study protocol (published)". This indicates that the protocol is already published. If it's already published, why is there a need of publishing the protocol again?

Authors’ response`: Agreed with the reviewer’s comments. Those statements were rephrased as follows

“A study protocol is developed following Preferred Reporting Items for Systematic Reviews and Meta-Analysis (PRISMA) guidelines [10]. The protocol is registered in the PROSPERO [11], International Prospective Register of Systematic Reviews with the registration number CRD42022357425 [12]”.

What geographical divisions will be sued by the authors?

Authors’ response: Here, 'geographical region' is nothing but the zone-wise division of Indian political territory.

Page 3, Line 97: What's the reason behind selecting the given timeline is not clear.

Authors’ response: In India, there was little work done before 2001. But, being more inclusive rather than exclusive, we like to add all the available relevant work in the present study.

Authors should consider writing methods section in paragraph rather than in bullet points. Strengths and limitations section needs to be more developed as well as paraphrased.

Authors’ response`: Agreed with the reviewer’s comments, we like to mention that we can happily do the task, recommended by the Reviewer. But we are here compelled to follow the conventional way of protocol writing.

---

## [Decision Letter · Decision Letter 1]

22 Feb 2023

Sexually Transmitted Infections among Key Populations in India: A Protocol for Systematic Review

PONE-D-22-31768R1

Dear Dr. Biswas,

We’re pleased to inform you that your manuscript has been judged scientifically suitable for publication and will be formally accepted for publication once it meets all outstanding technical requirements. 

Kind regards,

Addisu Melese Dagnaw, MSc

Academic Editor

PLOS ONE

Additional Editor Comments (optional):

Reviewers' comments:

Reviewer's Responses to Questions

**Comments to the Author**

1. Does the manuscript provide a valid rationale for the proposed study, with clearly identified and justified research questions?

Reviewer #1: Yes

Reviewer #2: Yes

2. Is the protocol technically sound and planned in a manner that will lead to a meaningful outcome and allow testing the stated hypotheses?

Reviewer #1: Yes

Reviewer #2: Yes

3. Is the methodology feasible and described in sufficient detail to allow the work to be replicable?

Reviewer #1: Yes

Reviewer #2: Yes

4. Have the authors described where all data underlying the findings will be made available when the study is complete?

Reviewer #1: Yes

Reviewer #2: Yes

5. Is the manuscript presented in an intelligible fashion and written in standard English?

Reviewer #1: Yes

Reviewer #2: Yes

6. Review Comments to the Author

You may also provide optional suggestions and comments to authors that they might find helpful in planning their study.

Reviewer #1: All comments were answered to satisfaction and the paper is now acceptable for publication.

Please still check all grammar carefully. For example, 'Discussion' not 'discussions'. The results of the systematic review will be interesting.

Reviewer #2: The authors have addressed all comments adequately. I have no further comments. The manuscript can be accepted.

7. PLOS authors have the option to publish the peer review history of their article (what does this mean?). If published, this will include your full peer review and any attached files.

Reviewer #1: **Yes: **Sylvia Bruisten

Reviewer #2: **Yes: **Ranadip Chowdhury

---

## [Editor Report · Acceptance letter]

3 Mar 2023

PONE-D-22-31768R1 

Sexually Transmitted Infections among Key Populations in India: A Protocol for Systematic Review 

Dear Dr. Biswas:

I'm pleased to inform you that your manuscript has been deemed suitable for publication in PLOS ONE. Congratulations! Your manuscript is now with our production department. 

Kind regards, 

on behalf of

Mr. Addisu Melese Dagnaw 

Academic Editor

PLOS ONE